# Understanding the Interaction Effects between Dietary Lipid Content and Rearing Temperature on Growth Performance, Feed Utilization, and Fat Deposition of Sea Bass (*Dicentrarchus labrax*)

**DOI:** 10.3390/ani11020392

**Published:** 2021-02-03

**Authors:** Lydia Katsika, Mario Huesca Flores, Yannis Kotzamanis, Alicia Estevez, Stavros Chatzifotis

**Affiliations:** 1Hellenic Centre for Marine Research (HCMR), Institute of Marine Biology, Biotechnology and Aquaculture (IMBBC), Hersonissos, 71003 Heraklion, Crete, Greece; katsikalydia@gmail.com (L.K.); mariohuescaflores@gmail.com (M.H.F.); jokotz@hcmr.gr (Y.K.); 2IRTA, Centre de Sant Carles de la Ràpita (IRTA-SCR), Aquaculture Program, Crta. PobleNou, km 5.5, 43540 Sant Carles de la Ràpita, Spain; alicia.estevez@irta.cat

**Keywords:** temperature, dietary lipids, fat deposition, growth performance, body composition

## Abstract

**Simple Summary:**

There is a growing need to use more efficient feed in fish farming. Designing a proper diet could potentially overcome even negative environmental impacts on the growth of farmed fish. It has been repeatedly stated that the change in temperature followed by the change of seasons can significantly affect the growth of fish. The aim of this work is to study the interaction of the diet (specifically the percentage of fat in it) with the change of temperature in the growth performance, feed utilization and fat deposition of Sea Bass (*Dicentrarchus labrax*). We used two different temperature regimes (starting at 23 °C and then changed to 17 °C and starting at 17 °C and then changed to 23 °C); fed one of the two commercial diets 16.5 and 20% lipids. We observed better growth rate and feed consumption when fish transferred to warmer water, but no diet-temperature interaction was observed. Different temperature regimes did not affect muscle or liver composition but the different fat diet content affected somatic indexes. In both temperature regimes, for a higher feed intake and body weight, a high fat diet is a better practice.

**Abstract:**

This study was conducted to elucidate the interaction effects of temperature and dietary lipid levels (2 × 2 factorial experiment) on the growth performance, muscle, and liver composition in adult farmed European sea bass (*Dicentrarchus labrax*). Two groups of fish (190 g; 60 fish per group) were distributed in 12 tanks in triplicates and kept at two different temperature regimes; one starting at 23 °C and then changed to 17 °C for 61 days, and the other starting at 17 °C and then changed to 23 °C for 39 days. Two commercial diets containing both ~44% crude protein but incorporating different dietary lipid levels, 16.5% (D16) and 20.0% (D20) (dry matter (DM)), were fed to the fish to apparent satiation; the type of diet fed to each fish group remained constant throughout the experiment. Final body weight, weight gain, and specific growth rate were significantly higher for the fish group held at 23 °C compared to the fish group at 17 °C (before the temperature changes), while the dietary fat content did not have any profound effect in both groups. Furthermore, the different temperature regimes did not affect muscle or liver composition, but, on the contrary, dietary lipids affected hepatosomatic, perivisceral fat, and visceral indexes. Feed conversion ratio and specific growth rate were not affected by the dietary lipid level. An interaction of temperature and dietary lipid content was observed in daily feed consumption (DFC) and final body weight (FBW).

## 1. Introduction

Most of the efforts in the farming of European sea bass (*Dicentrarchus labrax*) are focused on improving growth performance, reducing feed conversion ratio (FCR) and fat deposition, as well as using alternative ingredients to fish meal (FM) and fish oil (FO) in the diets for on growing. Several studies corroborate the importance and relation between amelioration of diet formulation/quality and confronting the aforementioned challenges [1]. In addition, fish body composition is an estimate of quality in fish farming. A higher amount of muscle over adipose tissue is a desirable aspect and satisfies the consumer impression of quality [2]. In fish species that do not store fat in their muscles, such as sea bass, even a small increase in muscle fat deposition can have strong impact on customer-perceived value [3,4]. This lipid accumulation is affected by the water temperature and dietary lipid content in an interrelated manner, which becomes more complex under seasonal water temperature changes [5]. It is common knowledge that an increase in water temperature can be beneficial for fish growth up to a certain level and may increase lipid accumulation [6].

Given the stressful situation caused by a substantial temperature change, emphasis has been placed on developing diets that aim to aid fish to cope with this challenge [7]. Lipids are the most important source of energy for fish growth, and the incorporation of well-balanced levels in fish diets can maximize protein sparing [8]. Furthermore, appropriate lipid levels in the feed provide the required essential fatty acids for the fish growth and improve the assimilation of fat-soluble vitamins [8,9]. The balance between protein and digestible energy must be taken into consideration. This is important because the protein content and its source have a direct impact on the cost of the feed [10], while high levels of non-protein energy sources may reduce feed intake, growth rates, and the quality of the fish fillet [1,11]. On the other hand, it is well known that the incorporation of fats or digestible carbohydrates (non-protein energy) up to a level into fish diets can improve protein retention and growth rates [12].

Fish are known to exploit protein preferentially to lipid or carbohydrate as an energy source [13]. According to the lipostatic regulation concept, fish are capable of adjusting the lipid stores to preserve their lipids at appropriate levels [14]. It has been shown that diet composition can significantly affect growth and adiposity in fish. The increase in dietary lipids has been found to improve growth in low-protein diets, but in the case of high protein diets (i.e., 53%), an increase in dietary lipids affects only the adipose tissue [15,16]. Dias et al. (2003) suggested that the optimal amount of dietary protein required for the maximum weight gain of 100–200 g European sea bass is 4–5 g kg^−1^ day^−1^ [17]. A fatty acid profile alteration could also have an impact on fillet’s quality, especially in terms of ω-3 HUFA composition [18].

Temperature is the environmental factor that affects feeding and fish growth to a great extent, and given its effect on fish metabolism, it also appears to affect the nutrient utilization efficiency [19]. The rhythms of metabolic processes in fish, as exothermic animals, are determined by temperature. At low temperatures and beyond their optimal developmental limits, fish slow down their metabolic rate by reducing the rate of energy intake [20]. In European sea bass, the temperature was also shown to affect feed efficiency and body composition [21]. The sea bass is an eurythermic species with optimal temperature for maximum growth of about 22 °C [22]. Generally, it has been shown that fish exhibit improved growth at higher temperatures within their tolerable limits, but temperatures higher than this range, known as “superoptimal temperature”, cause the opposite effects [23]. We can define optimal temperature as a range in which the energy retained for growth is maximum [16]. In aquaculture, the establishment of protocols with the optimum temperature for each species is vital as it leads to the most efficient fish growth, nutrient retention, survival, and fillet quality [24]. Seasonal changes in temperature can affect the digestibility of nutrients, for example, warmer water temperatures affect positively lipid digestibility [25]. In addition, water temperature can affect the ability of fish to digest and absorb lipids through changes in the composition of fatty acids [5]. Understanding the interaction between nutrients, especially dietary lipids, and the rearing temperature will provide information on the formulation of adequate diets that meet the nutritional requirements of fish throughout the whole grow-out period.

Considering the above, the aim of the present study was to investigate the effect of two dietary lipid levels (16.5% and 20.0%) on growth performance, muscle, and liver composition of European sea bass held under different seawater temperatures, which were designed to simulate a main cycle of changes that occur in Mediterranean autumn, winter, and spring (23→17 °C and 17→23 °C).

## 2. Materials and Methods

The trial was conducted at the Aqualabs facilities of the Institute of Marine Biology, Biotechnology and Aquaculture (IMBBC) of the Hellenic Centre for Marine Research (Heraklion, Greece), in 12 indoor circular black fiberglass tanks (470 l). The tanks were supplied with filtered borehole seawater (salinity 34‰) and oxygenated to saturation by air supply. Dissolved oxygen levels were at 7.7 mg/L and pH 7.6 and the photoperiod was 12/h light/12/h dark. The tanks were connected to a recirculation system and two groups of fish (60 fish/group) individually weighing ~190 g were randomly placed in each tank (20 fish/tank) and reared at two different temperature regimes (Group A starting at 23 °C and then changed to 17 °C and Group B starting at 17 °C and then changed to 23 °C); fed one of the two commercial diets containing 44% and 43% protein and 16.5 and 20% lipids (3 tanks × 2 temperatures × 2 lipid level). Fish were hand-fed to apparent satiation three times a day for 7 days a week. Uneaten pellets were siphoned out, dried, and weighted to estimate the correct amount of feed consumed by the fish. Ingredient and proximate composition of the experimental diets are presented in Table 1.

After the first 61 days of rearing, temperatures were reversed for both groups of fish (Group A fish kept at 23 °C were experienced 17 °C and Group B fish kept at 17 °C were experienced 23 °C), but fish continued to receive the same feeds as at the start of trial until day 100. Both at day 61 (end of the first test period), and at the end of the trial (day 100, end of the second test period), fish were starved for 24 h, lightly anesthetized, and their individual wet weight was recorded. Viscera and liver weights were recorded for the determination of hepatosomatic and visceral indexes and samples of liver and muscle were also collected from 3 fish per tank to analyze tissue composition. By the end of the trial, growth and somatometric indexes were determined. The growth parameters were calculated according to the following formula:

Feed Conversion Ratio (FCR) = feed intake/weight gain

Specific growth rate (SGR) (% body weight × day^−1^) = 100 × (ln (final body weight) − (ln initial body weight))/days

Hepatosomatic index (HSI) = 100 × weight of the liver/whole body weight

Perivisceral fat index (PFI) = (perivisceral fat/body weight) × 100

Viscerosomatic index (VSI) = 100 × weight of digestive tract/whole body weight

### 2.1. Biochemical Analyses

All the samples were stored at −20 °C, homogenized (Grindomix GM200 Retsch GmbH, Haan, Germany), and freeze-dried (freeze-drier Telstar Cryodos, Terrassa, Spain). Muscle samples were analyzed for dry matter (method 934.01) and ash (method 942.05) according to the work in [26]. The total moisture content was determined after drying the sample at 90 °C for about 4 h while for the ash content at 700 °C for 7 h in a muffle furnace (Heraeus D-6450 Hanau M110, Heraeus Instruments, Hanau, Germany). Crude lipids determined according to the work in [27]. Briefly, the fat was extracted using methanol/chloroform/ΒHΤ solution (2:1 methanol/chloroform *v*/*v* + 0.01% *w*/*v* BHT) (Sigma Aldrich, Darmstadt, Germany) in proportions 1:15 *w*/*v*. Crude protein was measured according to Dumas method (nitrogen content (N) × 6.25) using a nitrogen analyzer (FP-528, Leco Corporation, St. Joseph, MI, USA). Chemical analyses were carried out in triplicate.

### 2.2. Statistical Analysis

All data were tested for normality and homogeneity of variance prior to be subjected to one-way ANOVA using Kolmogorov–Smirnov and Levene’s tests, respectively. Factorial (two-way) analysis of variance (ANOVA) was used to determine the effects of temperature, dietary lipid levels, and their interactions on the growth performance, muscle composition, and liver composition, using SPSS 16.0 for Windows. Difference was considered as significance at *p* < 0.05. If two-way ANOVA interaction was significant, temperature and dietary lipid levels effects were analyzed by one-way ANOVA for each temperature and dietary lipid level, respectively.

### 2.3. Ethics Compliance

The experimental protocol was designed in accordance with the European Union Laws on the care and use of experimental animals (European directive 86 609/EEC) and the Greek codes of practice EL-91-BIO-03 and 04 for rearing and experimentation of marine organisms.

## 3. Results

The effects of dietary lipids and water temperature on the growth of European sea bass are shown in Figure 1 and Figure 2, and Table 1 and Table 2. After 61 days of feeding, the body weight and weight gain of the fish were significant higher for fish held at temperature 23 °C compared to counterparts kept at 17 °C for the same period (*p* < 0.001). At the end of the first test period (day 61), different levels of lipid content in diets had no effect on body weight and weight gain of the fish, regardless of the water temperature. At the end of the trial (day 100), body weight and weight gain were higher for fish transferred from 23 °C to 17 °C (Group A compared to Group B). It was also observed that final body weight was significantly higher for the fish fed the D20 diet than for those fed the D16 diet. Further, we did not observe any interaction between temperature and fat content in diet in final body weight and weight gain. At the end of the first period (day 61), SGR was higher in fish reared at 23 °C compared to fish reared at 17 °C (0.8% vs. 0.5% body weight per day). On the contrary, at the end of the trial (day 100), SGR in Group B (17 °C to 23 °C) increased rapidly and significantly exceeded that of Group A (23 °C to 17 °C). No effect of the two dietary lipid levels (D20, D16) was observed in the SGR for both experimental periods (day 61 and day 100). Therefore, dietary lipid level did not affect SGR, whereas temperature change significantly affected SGR (*p* < 0.001).

Feed conversion ratio (FCR) and feed consumption are shown in Figure 3 and Figure 4, and Table 3 and Table 4. In the first period (day 61), FCR was lower in Group A compared to Group B in the fish starting at 23 °C. At the end of the experiment (day 100), FCR did not differ between the two groups (Group A and Group B). More specifically, it was noticed that the FCR of Group A increased after transferring the fish to colder water, while the FCR of Group B improved when the fish were transferred to warmer water, resulting in similar FCR values for both groups. Our results showed a significantly higher daily feed consumption for Group A fish that were held at 23 °C during the first period (61 days). No significant impact of dietary lipid level was found on the daily feed consumption in both groups. At the end of the experiment, daily feed consumption was significantly higher in Group B compared to Group A, due to its transfer to warmer water. However, final daily feed consumption did not appear to be affected by the dietary lipid content in both groups (D20 or D16).

The effects of dietary lipid levels and water temperature on the somatic indexes (hepatosomatic, perivisceral fat, and viscerosomatic) of European sea bass are shown in Figure 5 and Table 5. Significantly lower somatic indexes for fish fed with D20 than those fed D16 were found for both groups A and B. The HSI differed among fish fed the same low-fat diet (D16); a lower HSI value (HSI = 1.6) was observed in those fish fed diet D16 and kept at 17 °C during the first phase of the trial (group B) compared to (HSI = 1.9) for those fish fed diet D16 and kept at 23 °C during the first phase of the trial (group A). For fish fed the same diet with high fat content (D20) and kept under different temperature regimes (either 17 or 23 °C), we did not notice differences in the values of HSI (1.4 vs. 1.3). The above results show an interaction between temperature and dietary lipid content in HSI; while for D20 the difference in temperature does not bring about any change, this does not apply for the for D16, where HSI is influenced by temperature. The perivisceral fat index and VSI were found higher for fish fed diet D16 regardless of the water temperature in both groups. In Group A, no significant difference was observed in the perivisceral fat index (3.8 vs. 4.6) between dietary lipid levels, but VSI was significant higher for fish fed D16 (VSI 7.9 vs. 9.9) in Group A. No statistically significant interaction between temperature and dietary lipid levels in perivisceral fat index or VSI was observed; overall, in any temperature regime (both group A and B), D20 shows lower values for both indices compared to D16.

The proximate composition of the muscle and liver of European sea bass held in two temperatures and fed diets containing different lipid levels are shown in Figure 6 and Figure 7. At the end of the trial, no differences were found in the composition of muscle and liver of the fish kept at different temperatures. Similarly, the lipid level of the diet did not significantly affect the proximate composition of both the muscle and the liver.

## 4. Discussion

Lipids are known to be an invaluable dietary energy source, and marine fish require them for optimal growth [28]. Several published results have suggested a close relationship between high dietary lipid content and fat deposition, although the tissue in which fat is accumulated is very different and depends on the species [12]. Thus, dietary lipid levels need to be carefully reviewed and defined, in order not to reduce the market value of the product. For the aquaculture industry, one of the main objectives of research [29] is the design of feeding protocols to minimize the feed waste without restricting fish growth and/or quality of the fillet. Mediterranean farmed fish do not live in a closed controlled environment. Environmental conditions change greatly throughout the year, with temperature being one of the main factors influencing their growth. Therefore, the knowledge of interaction of temperature with the feed composition is a powerful and innovative tool in the hands of fish producers. Precisely because temperature seems to affect the nutritional and physiological requirements of the cultivated species [5], the nutrition protocols must be developed and differentiated not only according to the species and their developmental stage, but also according to the environmental conditions (i.e., water temperature fluctuations). Nowadays, more attention is paid to the quality of the final product by consumers. Therefore, fish flesh composition and fat deposition are critical points for evaluating the final product [30].

Watanabe (1982) and Sheridan (1994) reported that muscle or liver composition (especially the lipid content) is not affected by dietary lipid levels as much as viscera or mesenteric fat content [31,32]. The results of the present study showed that the muscle and liver composition were not affected by dietary lipid levels, but HSI, PFI, and VSI were significantly lower for fish fed with high fat diet. The negative correlation of dietary fat with HSI has been reported before for the same species [33], but also for others [34,35]. In this short-term trial, we did not observe increased muscle adiposity for fish fed high fat diet. On the contrary, salmonids fed on diets that contained high lipid levels increased both their body fat deposition and VSI, the longer duration of this study (6 months) may have contributed to this apparent discrepancy between these results [36]. Other studies [3,37] reported that dietary lipids may cause fat deposition initially in the liver and peripheral adipose tissue and then in the muscle. El-Sayed et al. (1996) have shown that the decrease in water temperature significantly reduced the levels of proteins and lipids in the body of Nile tilapia (*Oreochromis niloticus*), which can be explained by the lower metabolic function that affects feed intake [38]. In our study, the temperature and dietary lipid regime affected HSI differently, fish fed the D16 diet and transferred from 23 °C to 17 °C (Group A) had higher HSI than those fed the same D16 and switch from 17 °C to 23 °C (Group B) but this was not observed in fish fed the high fat diet (D20), where HSI remained unchanged regardless of temperature regime. The HSI in the former treatment (Group A, D16) was higher despite the lower SGR, but without difference in liver fat deposition. According to Islam et al. (2020), an HSI increase could result from liver hyperplasia and hypertrophy [39] or in fish that. Although D16 was observed to be associated with higher HSI, liver composition analysis did not show increased liver fat deposition for this group. Note that PFI was significantly lower for fish fed with the high fat diet when they were transferred to warmer water (Group B). Perivisceral fat deposition is an index of the quality of the final product with high values corresponding to a poor quality fish [37,38,40].

It has been cited that excessive energy in the diets could limit fish growth as a result of a decrease in feed consumption [28]. The effect of temperature on the growth is related also to feed consumption and metabolism, as a significant amount of energy can be used to cover metabolic requirements instead of being utilized as body weight gain [41]. The results of the present study show the existence of a positive effect of high dietary no-protein energy content (D20) on FBW especially when fish transferred at warmer water temperature, in line with previous results in sea bass and other species [33,40], without any effect on FCR. However, Wang et al. (2005) found that juvenile cobia (*Rachycentron canadum*) fed diets containing 5% and 15% lipid showed a higher growth than those fish fed with 25% lipid [28]. This is often observable because there is an upper feeding limit for fish to utilize dietary lipids beyond which growth is declining due to low feed consumption values [42]. Bendiksen et al. (2003) observed that in Atlantic salmon (*Salmo salar*) an increase in dietary lipids resulted in a reduced weight gain at the highest temperature, whereas no such effect was observed at the lowest temperature [36]. Feed composition can significantly affect feed consumption when the fish are not fed to satiation but if the fish receive an adequate supply of feed other factors such as temperature, stomach size, and feed digestibility play an important role in feed consumption [43]. This agrees with our findings; fish were fed daily to satiation, but feed consumption was influenced only by temperature changes and not by the dietary lipids in both feeding periods (61 and 100 days) in both Groups A and B. However, in the first 61 days of the experiment, i.e., before the temperature change took place, within Group B the feed consumption was significantly higher when dietary fat content was high (D20). In fish of Group A, starting in 23 °C, the dietary fat content did not affect feed consumption after 61 days of feeding. A study by Person-Le Ruyet et al. (2004) concluded that Mediterranean sea bass populations can grow at an optimum rate only for a short time (2 months) at 25 °C and 29 °C if they are provided with the adequate concentration of oxygen and feed [21]. At the end of the present trial, the feed consumption of Group B fish was initially kept at 17 °C was approximately 60.4% of the consumption of Group A fish at 23 °C, and growth was significantly lower.

Although the increase in dietary fat for 16 to 20% improved growth, this growth was sufficient enough to compensate the reduction of growth cause by the decrease in rearing temperature from 23 to 17 °C. Nevertheless, we assert that feeding a high-fat diet to fish acclimatized to warmer waters is a good practice for improving growth. This agrees with previous results in this species [36,42] as well as in other species [39,43]. The decrease in FCR at the high temperature in Group A compared to Group B (during the first period) can be explained by the increase in feed intake which the fish needed to reach their growth inherent potential as the temperature was close to the optimum for this species. At the end of the second period (day 100), FCR of Group A (23 to 17 °C) increased while FCR of Group B (17 to 23 °C) decreased, although the differences in FCR between the two groups at day 100 were not statistically significant. The temperature for best feed consumption in many species, such as Nile tilapia (*Oreochromis niloticus*), hapuku (*Polyprionoxygeneios*), and others, is slightly higher than the one for best growth, and the temperature for best FCR is lower that the temperatures for best feed consumption and growth [44,45,46] This is in line with our results, as we found higher feed consumption and SGR in Group B (17–23 °C) without an associated improvement in FCR at the end of the trial (day 100).

## 5. Conclusions

Farmed European sea bass encounter temperature fluctuations during their grow-out period. Therefore, it is important to understand the interactions of temperature and dietary fat, and their effects on growth, body composition, and fat deposition in sea bass. According to the first period, fish reared at 23 °C showed higher final body weight, weight gain, feed consumption, and specific growth compared to fish reared at 17 °C; while dietary fat content did not have any profound effect on the above parameters, except for feed consumption. We observed higher feed consumption for fish fed D20 at 17 °C during the first period. By the end of the trial, FBW and weight gain remained higher for Group A (23 to 17 °C), but SGR and feed consumption were better for Group B (17 to 23 °C). Only in feed consumption (day 61) and body weight (day 100) an interaction was recorded between temperature and fat diet content. In both temperature regimes, for a higher DFC and FBW, a high fat diet is a more promising practice. In addition, the different temperature regimes did not affect muscle or liver composition but on the contrary the different dietary lipids affected hepatosomatic, perivisceral fat, and visceral indexes. The feed conversion ratio and specific growth rate were not affected by the dietary fat level at the end of the trial.

## Figures and Tables

**Figure 1 animals-11-00392-f001:**
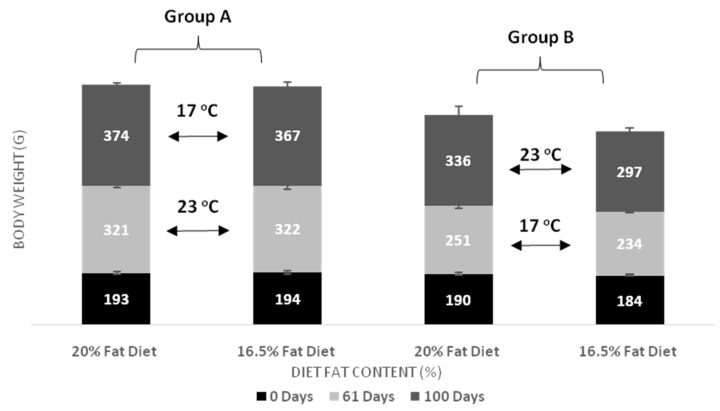
Effect of dietary lipid level and water temperature on body weight (g) of European sea bass (mean ± SD; *n* = 3).

**Figure 2 animals-11-00392-f002:**
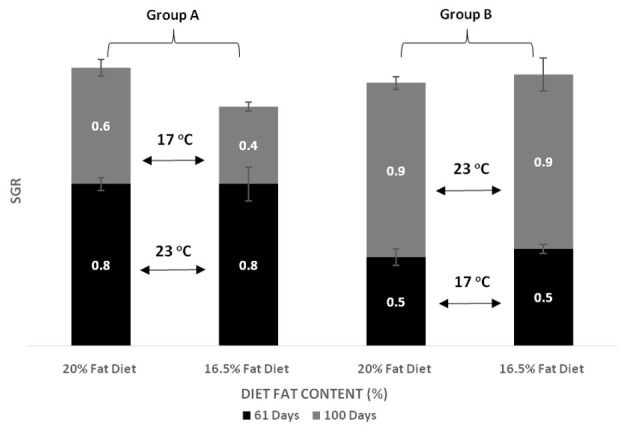
Effect of dietary lipids and water temperature on growth (SGR) of European sea bass (mean ± SD; *n* = 3).

**Figure 3 animals-11-00392-f003:**
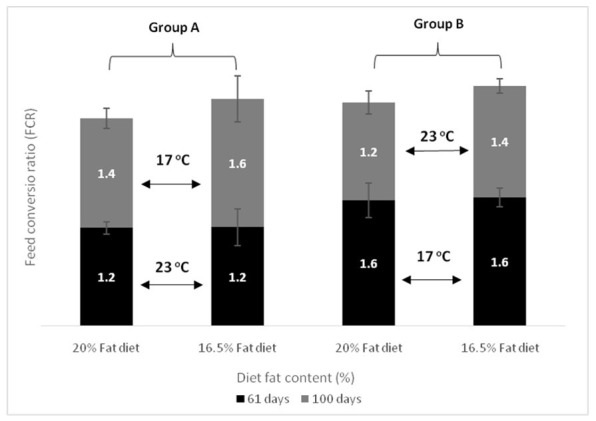
Effect of dietary lipids and water temperature on Feed Conversion Ratio (FCR) of European sea bass (mean ± SD; *n* = 3).

**Figure 4 animals-11-00392-f004:**
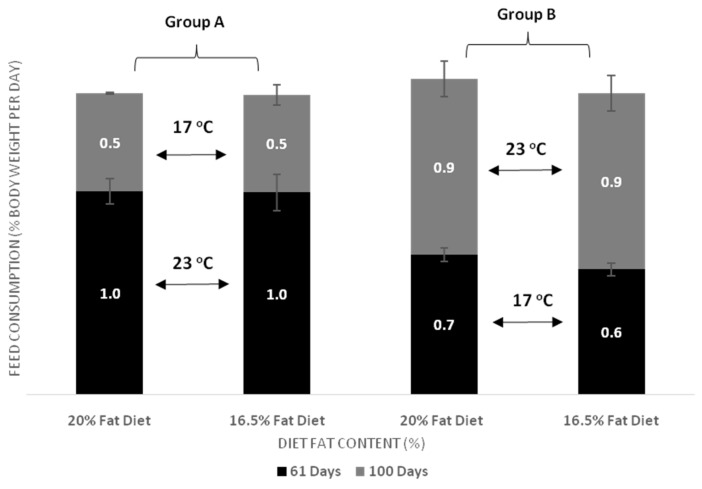
Effect of dietary lipids and water temperature on daily feed consumption of European sea bass (mean ± SD; *n* = 3).

**Figure 5 animals-11-00392-f005:**
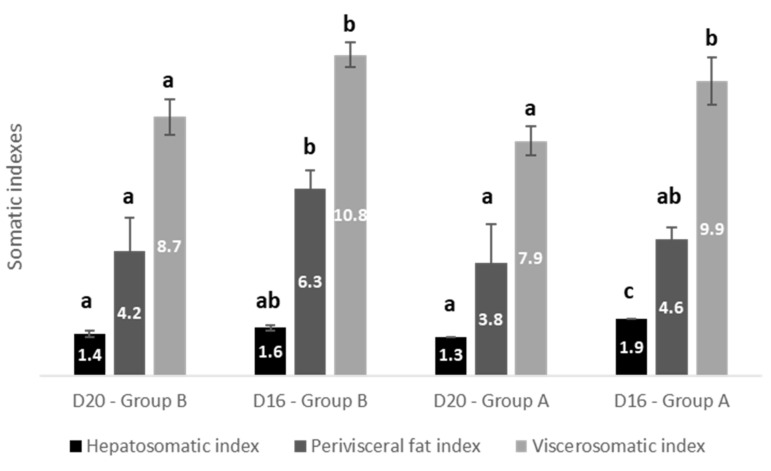
Effect of dietary lipids and water temperature on somatic indexes of European sea bass (%wet weight, mean ± SD *n* = 9). Different letters between columns denote statistically significant difference in indexes among the different dietary lipid levels and water temperature.

**Figure 6 animals-11-00392-f006:**
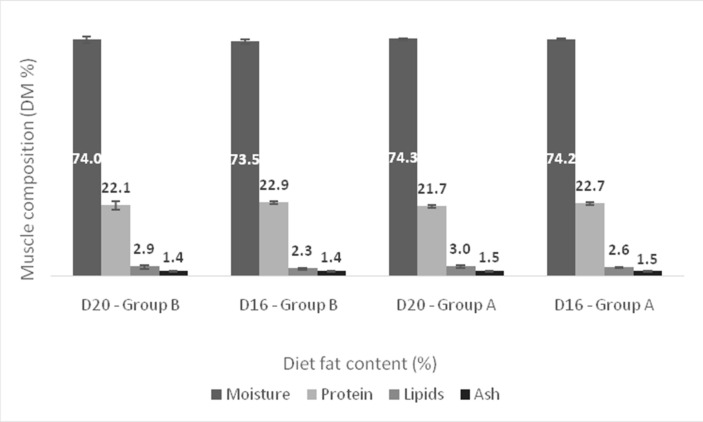
Effect of dietary lipids and water temperature on proximate composition of muscle (%dry weight, mean ± SD; *n* = 6). No statistically significant differences were observed.

**Figure 7 animals-11-00392-f007:**
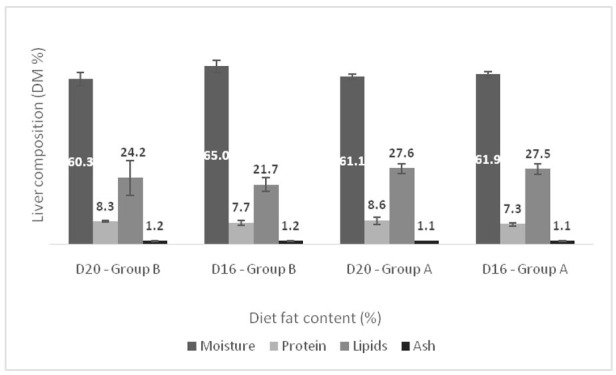
Effect of dietary lipids and water temperature on proximate composition of liver (% dry weight, mean ± SD; *n* = 6). No statistically significant differences were observed.

**Table 1 animals-11-00392-t001:** Composition and proximate analysis of the commercial experimental diets.

Ingredients (%).	D16	D20
Fish meal 67%	21.0	20.0
Fish oil	12.0	16.0
Wheat meal 44%	19.0	20.0
Soybean meal	7.5	8.0
Sunflower seed meal	-	2.0
Poultry meal	4.0	8.0
Hydrolyzed feather meal	17.0	13.5
Blood meal (pork/poultry)	5.0	-
Soy protein concentrate	12.0	10.0
Monocalcium phosphate	1.5	1.5
Proprietary amino acid–vitamin–mineral mix	1.0	1.0
**Total**	**100.0**	**100.0**
Proximate composition of the experimental diets (DM)		
Protein	44.0%	43.0%
Fat	16.5%	20.0%

**Table 2 animals-11-00392-t002:** Summary table of the two periods of the trial with the statistically significant differences in body weight and weight gain.

	Temperature	Dietary Fat Content	Interaction
Body Weight day 61	Group A vs. Group B, *p* < 0.001 Temperature in D20, *p* < 0.001 Temperature in D16, *p* < 0.001	*p* = 0.202	*p* = 0.152
Body Weight day 100	Group A vs. Group B, *p* < 0.001 Temperature in D20, *p* = 0.017 Temperature in D16, *p* = 0.002	D20 vs. D16, *p* = 0.02 Fat diet in Group B, *p* = 0.024	*p* = 0.323
Weight Gain day 61	Group A vs. Group B, *p* < 0.001 Temperature in D20, *p* < 0.001 Temperature in D16, *p* < 0.001	*p* = 0.356	*p* = 0.320
Weight Gain day 100	Group A vs. Group B, *p* = 0.023	*p* = 0.103	*p* = 0.929

**Table 3 animals-11-00392-t003:** Summary table of the two periods of the trial with the statistically significant differences in SGR.

	Temperature	Dietary Fat Content	Interaction
SGR day 61	Group A vs. Group B, *p* < 0.001 Temperature within D20, *p* < 0.001 Temperature within D16, *p* < 0.001	*p* = 0.341	*p* = 0.341
SGR day 100	Group A vs. Group B, *p* < 0.001 Temperature within D20, *p* < 0.001 Temperature within D16, *p* < 0.001	*p* = 0.719	*p* = 0.552

**Table 4 animals-11-00392-t004:** Summary table of the two period of the trial with the statistically significant differences in FCR and DFC.

	Temperature	Dietary Fat Content	Interaction
FCR day 61	Group A vs. Group B, *p* < 0.001 Temperature within D20, *p* = 0.002 Temperature within D16, *p* = 0.001	*p* = 0.771	*p* = 0.727
FCR day 100	*p* = 0.130	*p* = 0.083	*p* = 0.768
DFC day 61	Group A vs. Group B, *p* < 0.001 Temperature within D20, *p* < 0.001 Temperature within D16, *p* < 0.001	Fat diet within Group B, *p* = 0.031	*p* = 0.148
DFC day 100	Group A vs. Group B, *p* < 0.001 Temperature within D20, *p* = 0.002 Temperature within D16, *p* = 0.004	*p* = 0.588	*p* = 0.744

**Table 5 animals-11-00392-t005:** Summary table of the statistically significant differences in somatic indexes at the end of the trial.

	Temperature	Dietary Fat Content	Interaction
HSI	Temperature within D16, *p* = 0.014	D16 vs. D20, *p* < 0.001 Fat diet within Group A, *p* < 0.001	*p* = 0.017
Perivisceral fat	*p* = 0.061	D16 vs. D20, *p* = 0.022 Fat diet within Group B, *p* = 0.032	*p* = 0.423
VSI	*p* = 0.052	D16 vs. D20, *p* = 0.022 Fat diet within Group B, *p* = 0.005 Fat diet within Group A, *p* = 0.005	*p* = 0.955

## Data Availability

Not available.

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
