# Peer review of "Understanding the Interaction Effects between Dietary Lipid Content and Rearing Temperature on Growth Performance, Feed Utilization, and Fat Deposition of Sea Bass (*Dicentrarchus labrax*)"

_animals, 2021, doi:10.3390/ani11020392_

Round 1
Reviewer 1 Report
This study deals on the potential effect of two rearing temperature regimes on the utilization of dietary lipids by the European sea bass. The objectives were clearly stated and, overall, the experiment was correctly done though the experimental design can be improved. However, the way the data have been treated and the results have been exposed needs to be substantially improved. In the current form, the manuscript is difficult to understand.
Firstly, I strongly recommended to add the line numbers in the manuscript to make easier the reviewer’s comments.
Two basic temperature treatments were tested, one starting at 23 ˚C and then changed to 17 ˚C, and the other starting 17 ˚C and then changed to 23 ˚C. Both treatments have periods at 17 and 23 ˚C. However, in Results and Discussion these treatments are simply called as either 17 or 23 ˚C what is confusing for the reader. For instance, in the first paragraph of Results, in the sentence: ”Final body weight (FBW), weight gain, and specific growth rate (SGR) increased when the fish were reared at 23 ºC water temperature compared to 17 ºC”; I’m not sure if the authors are referring to differences between both temperature regimes (17 to 23˚C vs 23 to 17˚C) or between both periods within each treatment. The same in the second paragraph, “Neither dietary lipid levels nor water temperature affected FCR over the trial period”. If you are referring to the final result, in grey in Fig 3, this assertion is true, but if you are referring to the first period, in black in Fig. 3, we can read FCR 1.2 vs 1.6. On the contrary, in the next sentence “Feed consumption was significant higher when the fish were kept at high temperature without having any influence by the lipid dietary content”, it seems that here the higher value (0.9) of the total experiment (in grey in Fig. 4, right side) was associated to the temperature of the second period, because if you follow the criterion of Fig 1 (temperature of the first period), the total feed consumption is lower at the higher temperature (0.5 in grey in Fig. 4 left side). The presence or absence of significance is difficult to read due to the confusing structure of the graphs. Are the letters and asterisks included only for the comparison of results at the end of the experiments? Were the results of the first period not compared? The asterisk seems to indicate differences between lipid levels within each temperature treatment like the letters, or are they indicating differences between temperatures in each treatment?
In addition, all measurements have been obtained considering two periods; from the start of the experiment to the change of temperature (61 days), and from the start to the end of the experiment (100 days). Therefore, the second period (from temperature change to the end of the experiment; 39 days) was not separately considered and it should be, because the final result is the sum (or average) of what occurred in both periods.
Likewise, figures 1 to 4 are represented in a confusing way. I assume that left side graphs are 23 to 17 ˚C treatment, and right side are 17 to 23 ˚C, although it is not explained.
The figures 5 to 7 are more comprehensible because they explain the results obtained at the end of the experiments. Nevertheless, there is again confusion between water temperature and temperature regime along the experiment in the legends of figures and the text. The figures are not showing the effect of water temperature but the effect of two different regimes of temperatures.
The first paragraph of Discussion is repetitive with Introduction. This paragraph should be summarised focusing on what has been examined in this study.
The duration of the experiment should be consistent throughout the experiment: 101 days in Material and Methods, 100 days in Results, and 91 days in Discussion. Furthermore, the second period is shorter that the first one as commented above. This fact should be taken into account when comparing results. Discussion and Conclusions are based on final results (but being mainly related to the temperature of the first period). It is not correct to assert that a decrease or increase of temperature yields a given result. That result may change if the second period has a different duration, and the feeding and growth parameters were not separately examined for second period. Again, according to the experimental design, you are not comparing water temperatures but two specific regimes of temperatures.
In summary, the data, statistics, manuscript and graphs should be thoroughly reworked to explain the results and conclusions in a more comprehensive manner.
Author Response
Reviewer #1
1) Firstly, I stronglyrecommendedtoadd the linenumbers in the manuscripttomakeeasier the reviewer’scomments.
Answer: Done
2) Two basic temperature treatments were tested, one starting at 23 ˚C and then changed to 17 ˚C, and the other starting 17 ˚C and then changed to 23 ˚C. Both treatments have periods at 17 and 23 ˚C. However, in Results and Discussion these treatments are simply called as either 17 or 23 ˚C what is confusing for the reader. For instance, in the first paragraph of Results, in the sentence:” Final body weight (FBW), weight gain, and specific growth rate (SGR) increased when the fish were reared at 23 ºC water temperature compared to 17 ºC”; I’m not sure if the authors are referring to differences between both temperature regimes (17 to 23˚C vs 23 to 17˚C) or between both periods within each treatment.
Answer: Thank you very much for your comments on the presentation of the results. We completely understood the confusion caused by the section of the results and tried to render this section of the manuscript with more clarifications. We divided the presentation of the results in two periods for the figures 1-4. Specifically, we refer as the first period (from the beginning of the experiment until day 61) and as the second period the remaining 39 days until the end of the experiment (day 100). Also, in figures 1-4 we mention two groups; Group from 23 ºC to 17 ºC and group from 17 ºC to23 ºC. Therefore, in each figure we compare the values of the first period between the two groups (first and second black column VS third and fourth black column) and the values of the second period between the two groups (first and second light column VS third and fourth light grey column). In figure 1, first period has light grey columns and second period dark grey columns. So, in each period the two groups have same water temperature but different diets. In addition we compare the differences among same diets but different temperatures. Always separated periods, horizontal comparison of values from the first to the fourth column.
Because there would be confusion with the signage of the statistically significant differences for all the different comparisons in the chart, we considered that perhaps one solution is to place them in a separate table below each chart.
3) Neither dietary lipid levels nor water temperature affected FCR over the trial period”. If you are referring to the final result, in grey in Fig 3, this assertion is true, but if you are referring to the first period, in black in Fig. 3, we can read FCR 1.2 vs 1.6.
Answer: The manuscript section has been modified. A summary table of the two periods of the trial with the statistically significant differences in FCR has been completed.
4) On the contrary, in the next sentence “Feed consumption was significant higher when the fish were kept at high temperature without having any influence by the lipid dietary content”, it seems that here the higher value (0.9) of the total experiment (in grey in Fig. 4, right side) was associated to the temperature of the second period, because if you follow the criterion of Fig 1 (temperature of the first period), the total feed consumption is lower at the higher temperature (0.5 in grey in Fig. 4 left side).
Answer: The manuscript section has been modified. A summary table of the two periods of the trial with the statistically significant differences in feed consumption has been completed.
5) Are the letters and asterisks included only for the comparison of results at the end of the experiments?
Answer: Your comment on the signage was perfectly insightful as the differences were not clearly discernible. A summary table of the differences was included.
6) Were the results of the first period not compared?
Answer: Done
7) The asterisk seems to indicate differences between lipid levels within each temperature treatment like the letters, or are they indicating differences between temperatures in each treatment?
Answer: Modified
8) In addition, all measurements have been obtained considering two periods; from the start of the experiment to the change of temperature (61 days), and from the start to the end of the experiment (100 days). Therefore, the second period (from temperature change to the end of the experiment; 39 days) was not separately considered and it should be, because the final result is the sum (or average) of what occurred in both periods.
Answer: Modified
9) The first paragraph of Discussion is repetitive with Introduction. This paragraph should be summarised focusing on what has been examined in this study
Answer: It is true that there is a deliberate repetition of ideas, but we considered it right to include the existing knowledge and link it to the beginning of the discussion of the results.
10) The duration of the experiment should be consistent throughout the experiment: 101 days in Material and Methods, 100 days in Results, and 91 days in Discussion. Furthermore, the second period is shorter that the first one as commented above. This fact should be taken into account when comparing results.
Answer: Modified.

Reviewer 2 Report
Comments to the authors:
This study investigates the effect of temperature and dietary fat content on growth performance and body composition in European sea bass. Findings support that temperature is a significant regulator of growth while diet affected nutrient partitioning. This study is important for determining optimal husbandry and diet strategies that will improve aquaculture efficiency. However, there are several concerns that the authors should consider during the revision process.
Major concerns:
- The primary issue is that it is not clear what direct comparisons are being used to support the statements regarding main effects of temperature. The statistics section states that a 2-way anova was used to identify effects of temperature and diet. However, in this case the temperature effect is more accurately described as the “temperature program” (ie: the 23 to 17 C treatment vs the 23 to 17C treatment), especially when comparing response variables like final body weight, SGR, and FCR. It is not possible to isolate the effect of the individual temperatures (23 vs 17 C) without pair-wise comparisons of the response variables at the mid-point and end-point. It is not clear that these comparisons have been made.
- The stacked bar charts are very confusing and bar height does not seem proportional to the numbers provided. For example, the results state the FBW and weight gain increased when fish were reared at 23C. That seems accurate in Figure 1 when comparing the height of the light gray bars (0-61 d), but the greater height of the dark gray bars (62-101d) suggest that the 17C grew faster than the 23C. It is strongly suggested to present these data as a line graph to clearly indicate weight gain over time. Other response variables might also be better suited for this type of presentation.
- It is not clear what differences the letters and asterisks are indicating. Again, referring to figure 1: different letters indicate that the 20% fat diet outperforms the 16.5% diet. The data in Figure 1 for the 17 to 23 treatment supports higher FBW for the 20% fat. However, in the 23 to 17 treatment the body weight after the 23C is identical between diets (321 vs 322) and differs by less than 2% at the end of the study. These data do just not support a diet effect. In addition, the asterisk is used to denote significant differences between levels of temperature. Does this mean 17 vs 23 within the same bar (this would not be a good comparison because treatment length is different), 17 vs 23 across the same diet and time frame (more accurately determined by a pair-wise comparison), or simply among the different temperature programs?
- Figure 2 and 4: I have the same question with the temperature effect as I do in Figure 1. Specifically, it is not clear exactly what two (or four?) means are being compared to identify differences between levels of temperature. Indicating each mean as a separate entity on a bar graph would be helpful to draw these comparisons.
- I am equally confused by Figure 5. I assume we are looking at differences in, for example, the four HSI bars (black). If so, the data indicate that 1.5 and 1.7 are not different (they share an “a”), but 1.5 and 1.4 are different (a vs b). Letters on the other response variable also do not make sense with their numerical differences.
- In summary, I think there are two critical errors in the authors approach to presenting their data. 1) They present individual means from each temperature x diet combination, but associate the means with p-values from the two-way anova main effects. To remedy this, a one-way anova should be performed followed by a multiple means comparison. 2) A two-way anova performed on final response variables does not provide statistical evidence for differences between individual temperatures (23 vs 17). This can only be determined by analyzing responses at the mid-point and end-point of the study. Avoiding the stacked graphs may help clarify the direct comparisons that are being made to support temperature effects.
Minor issues:
- The statement in lines 37-39 requires a reference(s)
- Line 41: Sentence should be revised to indicate that a substantial temperature change cause stress, not that a stressful situation causes a temperature change as it is written
- It is assumed that fish were weighed at the beginning of the trial, at the temperature transition (day 61), and at the end of the trial. Line 106-108 indicates that fish were weighed only at the end of the trial. Please revise accordingly.
- Line 125: Identify the method of fat analysis (ether extraction, etc)
- Line 104: revered should be reversed
Author Response
Reviewer #2
- The primary issue is that it is not clear what direct comparisons are being used to support the statements regarding main effects of temperature. The statistics section states that a 2-way anova was used to identify effects of temperature and diet. However, in this case the temperature effect is more accurately described as the “temperature program” (ie: the 23 to 17 C treatment vs the 23 to 17C treatment), especially when comparing response variables like final body weight, SGR, and FCR. It is not possible to isolate the effect of the individual temperatures (23 vs 17 C) without pair-wise comparisons of the response variables at the mid-point and end-point. It is not clear that these comparisons have been made.
Answer: Thank you very much for your comments on the presentation of the results. We completely understood the confusion caused by the section of the results and tried to render this section of the manuscript with more clarifications. We divided the presentation of the results into two periods for the figures 1-4. Specifically, we refer as the first period (from the beginning of the experiment until day 61) and as the second period the remaining 39 days until the end of the experiment (day 100). Also, in figures 1-4 we mention two groups; Group from 23 ºC to 17 ºC and group from 17 ºC to23 ºC. Therefore, in each figure we compare the values of the first period between the two groups (first and second black column VS third and fourth black column) and the values of the second period between the two groups (first and second light column VS third and fourth light grey column). In figure 1, first period has light grey columns and second period dark grey columns. So, in each period the two groups have same water temperature but different diets. In addition we compare the differences among the same diets but different temperatures. Always separated periods, horizontal comparison of values from the first to the fourth column.
Because there would be confusion with the signage of the statistically significant differences for all the different comparisons in the chart, we considered that perhaps one solution is to place them in a separate table below each chart. All the Pairwise Multiple Comparison Procedures were done. Comparisons for factor: Temperature, Comparisons for factor: Fat diet, Comparisons for factor: Fat diet within 17 to 23, Comparisons for factor: Fat diet within 23 to 17, Comparisons for factor: Temperature within 20%fat, Comparisons for factor: Temperature within 16,5%fat All done.
- The stacked bar charts are very confusing and bar height does not seem proportional to the numbers provided. For example, the results state the FBW and weight gain increased when fish were reared at 23C. That seems accurate in Figure 1 when comparing the height of the light gray bars (0-61 d), but the greater height of the dark gray bars (62-101d) suggest that the 17C grew faster than the 23C. It is strongly suggested to present these data as a line graph to clearly indicate weight gain over time. Other response variables might also be better suited for this type of presentation.
Answer: The manuscript section has been modified. A summary table of the two periods of the trial with the statistically significant differences in FBW has been completed. We suggested this type of graph to show the values of each index in the columns for a quick visual estimation. It is true that the differences presented between the comparisons before were not understood so we suggest as a possible solution to improve the manuscript with the summary tables.
- It is not clear what differences the letters and asterisks are indicating. Again, referring to figure 1: different letters indicate that the 20% fat diet outperforms the 16.5% diet. The data in Figure 1 for the 17 to 23 treatment supports higher FBW for the 20% fat. However, in the 23 to 17 treatment the body weight after the 23C is identical between diets (321 vs 322) and differs by less than 2% at the end of the study. These data do just not support a diet effect. In addition, the asterisk is used to denote significant differences between levels of temperature. Does this mean 17 vs 23 within the same bar (this would not be a good comparison because treatment length is different), 17 vs 23 across the same diet and time frame (more accurately determined by a pair-wise comparison), or simply among the different temperature programs?
Answer: Your comment on the signage was perfectly insightful as the differences were not clearly discernible. A summary table of the differences was included .
- Figure 2 and 4: I have the same question with the temperature effect as I do in Figure 1. Specifically, it is not clear exactly what two (or four?) means are being compared to identify differences between levels of temperature. Indicating each mean as a separate entity on a bar graph would be helpful to draw these comparisons.
Answer: The manuscript section has been modified. We hope that after the correction of the text and the addition of the summary tables the presentation of the comparisons and differences became more understandable.
- I am equally confused by Figure 5. I assume we are looking at differences in, for example, the four HSI bars (black). If so, the data indicate that 1.5 and 1.7 are not different (they share an “a”), but 1.5 and 1.4 are different (a vs b). Letters on the other response variable also do not make sense with their numerical differences
Answer: It is right that we are looking at differences in, for example, the four HSI bars (black). In summary tables we also quote the p values. 1.6 and 1.9 are different, 1.4 + 1.3 different from 1.6 + 1.9, and the last one 1.3 and 1.9 are different.
- The statement in lines 37-39 requires a reference(s)
Answer: Done
- Line 41: Sentence should be revised to indicate that a substantial temperature change cause stress, not that a stressful situation causes a temperature change as it is written
Answer: Modified
- It is assumed that fish were weighed at the beginning of the trial, at the temperature transition (day 61), and at the end of the trial. Line 106-108 indicates that fish were weighed only at the end of the trial. Please revise accordingly.
Answer: Modified
- Line 125: Identify the method of fat analysis (ether extraction, etc)
Answer: Done
- Line 104: revered should be reversed
Answer: Done

Round 2
Reviewer 1 Report
The authors made an important effort in improving the manuscript. The revised version is fine with me. I have no further comments.
Author Response
Dear Sir,
We have replied to your comments in the file attached.

Reviewer 2 Report
The revised manuscript is improved compared to the initial submission, particularly with the inclusion of tables describing the treatment comparisons. As a result, there is better statistical support for the findings and conclusions.
My only outstanding issue is with the lettering in Figure 5. If treatment effects are only detected using the comparisons in Table 5 then the graph should not have letters above each graph. Individual letters, however, are appropriate if a multiple means comparison analysis was performed. In this case, a multiple means comparison is appropriate since there was an interaction between independent variables, at least for HSI. P
- HSI: Currently letters indicate that 1.4, 1.3, and 1.6 are all similar (they share an "a"). Should the "a" be dropped from 1.6?
- PFI: Letters indicate that 3.8 and 4.6 are similar but 4.2 and 4.6 are different. This does not seem correct. Table 5 also states there is a fat diet effect within group B, so 6.3 and 4.6 differ. However, they share a "c", supporting similar means.
Please ensure consistency between the lettering in Figure 5 and the significant differences reported in Table 5.
Author Response
Dear We have replied to your comments in the file attached.
